# An Introduction to AI for GLAM

**Daniel van Strien** [* 1]  **Mark Bell** [* 2]  **Nora McGregor** [* 1]  **Michael Trizna** [* 3]

## Abstract

There is a growing interest in utilising Machine Learning (ML) techniques within Galleries, Libraries, Archives and Museums (GLAM), and a corresponding demand for training to enable practitioners to engage confidently in this area. Staff at these institutions are seeking practical knowledge and skills in ML concepts and methods specific to the sector's work, such as in the curation and collection of heritage collections. In this paper, we discuss the motivations and methods behind 'An Introduction to AI for GLAM', a new Carpentries[1] workshop under development through an international partnership between British Library, Smithsonian Institution, and The National Archives UK. This new workshop aims to introduce GLAM practitioners to the essential conceptual and practical considerations for supporting, participating in and undertaking machine learning-based research and projects within the GLAM sector.

## 1. Introduction

The past decade has seen a growing exploration of Machine Learning (ML) by the Galleries, Libraries, Archives and Museums (GLAM) sector. This interest is reflected in a growing number of networking initiatives[2], research projects applying ML to GLAM collections (Lee et al., 2020; Lincoln et al., 2020),[3] and projects developing ML tools

---

[*]Equal contribution [1]British Library, London, UK [2]The National Archives, London, UK [3]Smithsonian Institution, Washington, DC, USA. Correspondence to: Daniel van Strien <Daniel.VanStrien@bl.uk>.

*Proceedings of the $2^{nd}$ Teaching in Machine Learning Workshop*, PMLR, 2021. Copyright 2021 by the author(s).

[1]https://carpentries.org

[2]These include: https://sites.google.com/view/ai4lam, https://www.cenl.org/networkgroups/ai-in-libraries-network-group/ and https://www.aeolian-network.net/

[3]In progress projects include; https://tanc-ahrc.github.io/DeepDiscoveries and https://livingwithmachines.ac.uk

aimed explicitly at GLAM institutions e.g. (Kahle et al., 2017).

A potential barrier to the effective adoption of ML more widely across the sector, lays in the difficulty of defining the precise skills set required for such transformation, when the potential role and opportunities for GLAM staff to engage in ML are so varied, and even, still up for debate (Cox, 2021). Should staff be directly building and training models? Should they be able to document training data developed from library content? Should they be able to work as part of a team developing ML models? While these different roles have different training requirements, they all require a basic understanding of ML. This paper briefly provides some background on the GLAM sector and existing training initiatives aimed at GLAM staff. It then introduces our Carpentries workshop, 'An Introduction to AI for GLAM'.

## 2. Machine Learning and the GLAM Sector

The GLAM sector encompasses a wide range of unique institutions whose staffing, budget, collections and primary audiences will necessarily differ in scale and scope. Though it is difficult to make broad statements about such a diverse sector that will be completely accurate, there are some areas of activity and focus common to all to some extent:

- Cataloguing and other forms of metadata generation.

- Enabling search and discovery of collections.

- Supporting and carrying out research.

- Public engagement and crowdsourcing.

These areas are all ones in which ML could be – or already is – having an impact. Further, the sector has a history of undertaking productive cross-collaboration amongst its members, with galleries, libraries, archives and museums benefiting from opportunities to share their distinct expertise and knowledge with each other (Zorich et al., 2008).This suggests that developing teaching material for the sector as a whole is a worthwhile approach.

## 3. GLAM focused ML Training Initiatives

GLAM institutions, including the British Library, The National Archives UK and the Smithsonian Institution, have all undertaken to develop a wide range of training opportunities for their staff in this area, taking various training approaches in getting this complex material across to a diversity of staff.

### 3.1. Library Carpentry

Library Carpentry is an initiative under the Carpentries umbrella focused on teaching software skills to library and information related communities. Library Carpentry was originally developed in 2014 in response to demand for access to software and data skills amongst people working in libraries (Baker et al., 2016). The Library Carpentry curriculum covers materials that would be relevant to GLAM staff wanting to work with ML, including introductions to 'tidy data', version control systems, use of the command line, and basic programming in Python. [4]

### 3.2. Digital Scholarship Training Programme at British Library

The Digital Research and Curator Team at the National Library of the UK, has run a digital and data skills training programme for British Library staff since 2012.[5] This programme creates opportunities for colleagues to develop necessary skills and knowledge to support, and undertake in their own right, emerging areas of modern scholarship such as the Digital Humanities. The programme supports several different modes of training delivering, from one-hour lunchtime lectures and a monthly reading group, 2 hour hands-on exploratory 'Hack and Yacks', to full day or week long courses on a given digital topic. The topic of ML has featured heavily throughout this programme in recent years, with a particular emphasis on introducing high level concepts and use cases in the application of ML in the GLAM sector to a novice audience, rather than delivering the software skills to implement them.

### 3.3. Machine Learning Club at The National Archives

The National Archives have run a number of activities under the banner of 'Machine Learning Club' for all interested staff. The initiative began with a one-off introductory talk, expanding to a series of talks covering multiple aspects of the topic. Enthusiasm amongst attendees to learn more led to the running of a series of workshops for technical and non-technical staff alike to gain hands-on experience of ML. As with the British Library, the aim was to teach concepts and for participants to understand their role within an ML ecosystem.

### 3.4. Computing for Cultural Heritage

The Computing for Cultural Heritage project was an Institute of Coding funded trial (2019/2021) that saw Birkbeck University of London, British Library and The National Archives UK develop a new one-year postgraduate certificate aimed at providing information professionals across GLAM sector with an understanding of basic programming, analytic tools and computing environments to support them in their daily work.[6] It was born out of a need to further support those who, having gained a keen interest in ML through institutional staff training, required the programming skills to practically undertake it.[7]

These diverse training initiatives, all designed from a GLAM practitioners perspective on ML, have had positive impacts, with participants reporting increased confidence in interactions with external data scientists, and being able to bring ML into their own research.[8]

## 4. An Introduction to AI for GLAM

In this section, we introduce our workshop, 'An Introduction to AI for GLAM', currently under development as part of the 'Carpentries Incubator'.[9] This next section outlines the aims, topics and delivery methods of this workshop alongside a reflection on why these were chosen.

The material for this course were developed as part of a Carpentries Lesson Development Study Group.[10] This study group took place over a couple of months and was intended to help participants develop new Carpentries lessons. Four members of the group, representing three GLAM institutions, decided to collaborate on this AI lesson.

---

[4]https://librarycarpentry.org/lessons/

[5]https://www.bl.uk/projects/digital-scholarship-training-programme

[6]https://www.bl.uk/projects/computingculturalheritage

[7]https://www.bl.uk/case-studies/computing-for-cultural-heritage-student-projects

[8]See for instance student projects undertaken by Computing for Cultural Heritage https://www.bl.uk/case-studies/computing-for-cultural-heritage-student-projects, https://blog.nationalarchives.gov.uk/computing-cholera-topic-modelling-the-catalogue-entries-of-the-general-board-of-health

[9]The incubator is a place for lessons to be developed outside of the core Carpentries curriculum https://github.com/carpentries-incubator

[10]https://carpentries-incubator.github.io/study-groups/

## 4.1. Learner Profiles

As part of the process of developing the lesson materials, learner profiles were created. The Carpentries recommend the development of learner profiles as a way of better identifying the target audience and their needs (Wilson, 2019). Learner profiles require the description of some characteristics of an expected learner for the material. For example; "What is their expected educational level?", "What type of exposure do they have to the technologies you plan to teach?" and "What are the pain points they are currently experiencing?" (Wilson, 2019). Although the people depicted in the profiles are fictional, the development of the profiles often drew on real people who work in our home institutions.

One challenge of using the learner profiles for our lesson was adapting them to focus slightly less on their technical skills. The learner profiles were also adapted slightly to focus more on our learners' potential attitudes toward ML. Whilst we were developing the lessons, we wanted to keep in mind the varying levels of enthusiasm for machine learning from the target audience.

## 4.2. Aims

The role that GLAM staff should play in ML projects remains an open question. Some GLAM institutions might choose to outsource most of their ML efforts to commercial solutions, whilst others will want their staff to be involved more directly. Whilst the stakes may not seem to be as high, this mirrors discussions in other disciplines, such as medicine, around the role domain experts should play and, crucially, what they need to know about ML (Sim et al., 2021; Olczak et al., 2021). We would argue that regardless of whether GLAM staff will be 'directly' involved, ML methods will be underpinning so many technologies in the future that basic literacy around ML is crucial for all GLAM staff.

Following this desire to develop a basic understanding of ML, 'An Introduction to AI for GLAM' has several aims. The primary overarching goal of the material is to provide an accessible introduction to ML for GLAM staff that is relevant to their work. In particular, the materials aim to introduce basic ML concepts and demystify training ML models. The material also seeks to provide a high-level overview of what ML is good at and where its limitations lie. Giving a realistic account of the field is especially important in the context of the growing use of ML in the GLAM sector and the risks of ML approaches being 'oversold' to staff.

Another central aim of the course is to emphasise ethical considerations. There are growing calls and examples of integrating ethics into ML course curricula (Garrett et al., 2020; Saltz et al., 2019). Highlighting potential ethical issues, particularly related to GLAM 'data' challenges, is a possible role for GLAM staff in ML (Coleman, 2020).

Finally, the lesson material aims to give a sense of the steps involved in a ML project, from identifying a 'business need' to deploying and monitoring models. The overview of these steps aims to help prepare GLAM staff to work as part of a ML project team rather than on the technical implementation of these different stages of a ML project.

## 4.3. Lesson Topics

We chose the topics covered in the workshop material with the learners' and aims in mind. The topics, as a result, differ from what may often be covered as part of an introductory ML course.

### 4.3.1. WHAT ARE ARTIFICIAL INTELLIGENCE AND MACHINE LEARNING?

This episode provides a high-level introduction to key concepts for understanding ML. In particular, it establishes the relationship between Artificial Intelligence and ML, distinguishes different types of ML and provides an overview of learning from data.

### 4.3.2. WHAT IS MACHINE LEARNING GOOD AT?

This episode provides an overview of tasks for which ML has shown good performance, with a particular focus on how these might be relevant in a GLAM setting. This episode also discusses some of the potential downsides of using ML.

### 4.3.3. UNDERSTANDING AND MANAGING BIAS

This lesson introduces different types of bias in ML and where these can be introduced in the overall pipeline and encourages a reflection amongst attendees on how GLAM staff can help manage this bias.

### 4.3.4. APPLYING MACHINE LEARNING

This episode looks at the broader process of carrying out a ML project and some of the considerations which will need to be made at different stages. It also discusses the potential roles GLAM staff can play in ML projects.

### 4.3.5. THE MACHINE LEARNING ECOSYSTEM

This episode will provide an overview of the current ecosystem, including a discussion of existing (benchmark) datasets, software libraries, and other tools for ML. It will also discuss the role of GLAM staff in this ecosystem, with a particular focus on the curation and documentation of models and data. This episode will also direct students to resources that will help expand their hands-on experience of machine learning for students interested in pursuing the topic further.

### 4.4. Delivery Methods

Unlike many introductory ML courses, our lessons do not include programming components.[11] However, the goal of the lesson material is still to be practically focused and interactive. The Carpentries place a strong emphasis on interactivity and the inclusion of exercises in lesson materials.

Since the lessons do not cover coding, we did not include coding exercises. Instead, the aims of the exercises were to:

- Test and develop an understanding of important ML concepts.

- Encourage a reflection of how ML could be used within a GLAM setting.

- Encourage a reflection on how ML could be utilised within specific institutions in which the participants of the workshops are based.

- Emphasise a critical reflection on the ethical issues that can be raised by using ML and particularly how these appear in the context of GLAM institutions and collections.

Examples of the types of exercises include;

- Multiple choice questions designed to test that concepts have been understood. For example, understanding the difference between supervised and unsupervised learning.

- A Discussion prompt for thinking about how ML could be utilised: 'How might object detection help speed up digitisation?'.

- Group discussion of 'points at which bias may enter the pipeline, and questions/strategies GLAM staff might want to consider in order to manage it.'

- A 'hands-on' activity exploring commercial computer vision services to reflect on their potential strengths and weaknesses.

We aimed for these exercises to build on experiences of running Library Carpentry workshops which includes similar discussion exercises.[12] These types of exercise are often not focused on teaching a specific technical competency i.e. how to write a Bash for-loop, but instead focus on increasing

the learners broad understanding of a topic and, crucially, their confidence. Whilst these exercises are not 'hands-on' in the traditional sense, they aim to ensure learners are actively engaged in the sessions. Our hope is that following the lesson, learners are able to apply their existing domain expertise in combination with what has been explored in the lesson and make a positive engagement with existing or proposed ML applications in their workplace.

### 4.5. Community Development and Maintenance

We have several reasons for developing this workshop as part of the Carpentries ecosystem. The Carpentries workshop materials are generated from source files in a GitHub repository. This allows for anyone to make a pull request or open an issue related to the lesson. Beyond this technical ability to make changes, the Carpentries organise regular events to encourage review and development of existing materials.

ML is a rapidly developing field with regular technical advances. Beyond this, there is also a growing maturity around the deployment of ML models in various domains. The use of ML in the GLAM sector will continue to develop over time, making it likely that aspects of our lesson will need to be updated. We hope that integrating the lesson materials inside the Carpentries ecosystem will help ensure that a broader community can update the material.

## 5. Conclusion

Introductory ML training that is grounded in the specific applications and use cases relevant to cultural heritage, that is practical, without being too overtly technical, will be key to ensuring the wider adoption of ML methods across GLAM. Though major cultural heritage institutions have undertaken in recent years to provide their own staff with a variety of training in this area, there is much to be gained by pooling expertise, resources, and experience to deliver a variety of open training materials available sector-wide. 'An Introduction to AI for GLAM' represents one effort, aiming to meet the growing demand for ML training specific to the sector, and provide a strong foundation for staff to gain confidence in entering this complex area. The Carpentries, and specifically Library Carpentry, though tending towards more technical lessons historically, provides the ideal home for such training. Library Carpentry acts as a natural signpost for digital skill seeking GLAM professionals, and has a diverse and dedicated community at the ready to ensure not just the open, shared and continued maintenance of the materials, but the development of related and more advanced courses branching off the core foundation as necessary.

---

[11]A notable exception is the Elements of AI course which had the goal of education 1% of European citizens in the basics of AI and doesn't use coding during the course https://www.elementsofai.com/faq/is-there-any-math-in-the-course

[12]An example includes a discussion exercise focused on 'jargon-busting' terminology used around code or software development.

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
