# OpenReview forum: "An Introduction to AI for GLAM"
_ecmlpkdd.org/ECMLPKDD/2021/Workshop/TeachML — TeachML 2021_

### Official Review · Reviewer_Lxmu · 2021-07-06
**Conceptual ML Teaching workshop for GLAM employees**

**Rating:** 6
**Confidence:** 5

**Review:**

Pros:
+ the idea developing a workshop to advocate/promote ML education for staffs working within Galleries, Libraries, Archives and Museums (GLAM) is helpful to a wide community.
+ detailed lesson topics, and student performance evaluation schemes are provided.
+ the introduction of ML ethics in the workshop.

Cons:
- typo in abstract: utilisation --> utilization
- the article would be stringer if it includes evaluation results (maybe through the form of a survey) to showcase the success of the workshop to promote ML education for staffs in GLAM.
- the workshop will be more beneficial to the employees if hands-on components are included. It may not necessarily be coding, but teaching the usage of pre-packaged ML tool will help them apply the learned concepts in their specific scenarios.

---

### Official Review · Reviewer_c26L · 2021-07-14
**This paper introduces learning material produced to introduce GLAM staff to Machine Learning.**

**Rating:** 8
**Confidence:** 5

**Review:**

pros
addessing an audience highly relevant in the realm of ML, because they provide data
well reviewed state of the field
well thought out pedagogical concepts, including delivery methods
thorough analysis of target audience needs
community oriented approach to lesson development and teaching

cons
provide a little more detail about the individual lesson components
explicitly include data curation and its relevance to machine learning

---

### Decision · Program_Chairs · 2021-07-21

**Decision:**

Accept

**Comment:**

Congratulations! The reviewers agree that this paper should be accepted.

Camera-ready version is due August 18, 2021. As you prepare the camera ready version, please take the reviewers comments into consideration.

We look forward to your participation at the workshop on September 13, 2021. We invite you also to join us for the satellite event on September 08, 2021. Schedules for both the workshop and the satellite event will be forthcoming.